# Whole-Person, Urobiome-Centric Therapy for Uncomplicated Urinary Tract Infection

**DOI:** 10.3390/antibiotics11020218

**Published:** 2022-02-09

**Authors:** Luciano Garofalo, Claudia Nakama, Douglas Hanes, Heather Zwickey

**Affiliations:** 1Department of Child, Family, and Population Health Nursing, University of Washington, Seattle, WA 98195, USA; 2National University of Natural Medicine, Portland, OR 97201, USA; claudia.young@nunm.edu (C.N.); dhanes@nunm.edu (D.H.); hzwickey@nunm.edu (H.Z.); 3Helfgott Research Institute, NUNM, Portland, OR 97201, USA

**Keywords:** UTI, cystitis, dysbiosis, microbiome, microbiota, urinary tract, urinary tract infection, urobiome, vaginal microbiome, alternative medicine

## Abstract

A healthy urinary tract contains a variety of microbes resulting in a diverse urobiome. Urobiome dysbiosis, defined as an imbalance in the microbial composition in the microenvironments along the urinary tract, is found in women with uncomplicated urinary tract infection (UTI). Historically, antibiotics have been used to address UTI. An alternative approach to uncomplicated UTI is warranted as the current paradigm fails to take urobiome dysbiosis into account and contributes to the communal problem of resistance. A whole-person, multi-modal approach that addresses vaginal and urinary tract dysbiosis may be more effective in reducing recurrent UTI. In this review, we discuss strategies that include reducing pathogenic bacteria while supporting commensal urogenital bacteria, encouraging diuresis, maintaining optimal pH levels, and reducing inflammation. Strategies for future research are suggested.

## 1. Introduction

Each year approximately 97 million outpatient visits are associated with antibiotic prescriptions, more of which are given for urinary tract infection (UTI) than any other diagnosis [1,2]. Urine culture and sensitivity testing are not recommended as the standard of care in the clinical diagnosis of UTI [3], and the management protocol is often irrespective of any clearly identifiable pathogen. Research has shown that antibiotic use is associated with an increased risk of recurrent UTI [4]. Moreover, antibiotic use within the past six months is a contributing risk factor for harboring resistant uropathogen isolates [5] and can facilitate cross-resistance when prescribed alongside non-antibiotics [6].

Trimethoprim–sulfamethoxazole (TMP–SMX) was recommended as first-line therapy in the treatment of acute uncomplicated bacterial cystitis by the Infectious Diseases Society of America guideline released in 2011 [7]. However, as TMP–SMX resistance has since escalated to a pooled worldwide average of 23.6% [8], providers may resort more to fluoroquinolones, causing an increase in the incidence of fluoroquinolone resistance among uropathogens, including ciprofloxacin-resistant *Escherichia coli* [9]. Fluoroquinolone resistance begets increasingly more dangerous infections such as hospital-acquired pneumonia and complicated UTI [10]. Exposure to fluoroquinolones has been associated with a more than 6-fold increased risk of acquiring *C. difficile* colitis [11]. Comparatively, patients who received TMP–SMX had a 2-fold higher risk of *C. difficile* colitis [11]. As the rates of incidence, recurrence, and therapeutic resistance of UTI rise, UTI treatment recommendations continue to evolve as part of the wider antibiotic stewardship effort.

Investigations into the efficacy of alternative therapies such as botanicals and probiotics have mostly been reduction-based, focusing on the impact of the substitute intervention on a singular physiological system in isolation. Many botanicals possess active constituents with antimicrobial properties or anti-adhesive effects that can aid in the treatment of UTI; however, more attention should be given to the complex mechanisms of action of complementary therapies as an integrative approach to Whole-Person Health (WPH).

This review applies a framework of WPH [12] to cystitis by examining the dynamic interactions between multiple organ systems and the microbiome that manifest as cystitis. First, we weave this holistic discussion of cystitis pathophysiology with the urgency of antibiotic resistance, amplifying the call for a new primary treatment strategy for uncomplicated cystitis. We then summarize the clinical evidence for nutritional, phytochemical, and behavioral therapies for UTI, examining their potential for symptom amelioration and restoring urobiome health. We conclude that a multi-modal approach consisting of behavioral changes and non-pharmaceutical products could be a safe and effective first-line approach towards the management of uncomplicated UTI.

### 1.1. Prevalence

UTI accounts for upwards of 10.5 million visits to physician offices and emergency departments per year and represents a significant portion of ambulatory healthcare costs [13]. UTI disproportionately affects women, with half of all women self-reporting at least one UTI by age 32 and 20–30% of women experiencing a second UTI within six months of initial infection [14]. These estimates do not include asymptomatic bacteriuria (ABU), occurring in 1% of schoolgirls, ≥2% of pregnant women, and about 20% of elderly individuals of both sexes [15]. Asymptomatic bacteriuria (ASB) is defined as the presence of ≥105 colony-forming units per mL of one or more bacterial species, irrespective of pyuria, in a urine specimen from a patient without signs or symptoms of UTI [16].

### 1.2. Pathophysiology

Infection typically begins when coliform bacteria contaminate the periurethral area and colonize the urethra through the expression of virulence and adhesion factors. From there, infection ascends to the bladder where bacterial multiplication is accomplished through the release of toxins and proteases that feed off host nutrients and instigate an inflammatory response [17]. By evading host immune surveillance, uropathogens can subsequently ascend to the kidneys, again attaching via adhesins or pili to colonize the renal epithelium and then produce tissue-damaging toxins [18].

### 1.3. Diagnostic Criteria

UTIs are categorized as either uncomplicated (simple) or complicated. Simple UTI, or simple cystitis, occurs primarily in otherwise healthy females of childbearing age. The prominent risk factors are prior UTI, new sexual partners (within the last year), vaginal infection, prolonged withholding of urine, and use of diaphragms or spermicides [19]. Complicated UTI is characterized by factors that disturb normal structure and function of the urinary tract, including indwelling catheters, anatomic variations or history of surgery in the urinary tract, renal insufficiency, neurogenic bladder, diabetes, pregnancy, and immunocompromised state (drug-induced, inherited, etc.). UTI in males is much less common and generally categorized as complicated [20]; similar to most of the literature on uncomplicated UTI, this review is directed toward UTI in females.

### 1.4. Cystitis as a Spectrum of Etiologies

Lower urinary tract symptoms can present similarly for a variety of non-infectious urological disorders as well as UTI. Chronic bladder inflammation is subject to central sensitization and can have a variety of host- or environmentally mediated triggers [21]. For this reason, it can be difficult to distinguish bacterial infection from interstitial cystitis or other non-infectious etiologies in the primary care setting, especially for UTI that is considered recurrent or chronic. As a result, many patients with cystitis of non-infectious etiology receive unnecessary antibiotic prescriptions.

Bacteriuria in the absence of symptoms does not justify antibiotic prescription unless in a pregnant female [22]. Uropathogenic bacteria are found with increasing frequency in the urine as patients age due to compounding factors such as menopause, immunosenescence, and recurrent colonization by drug-resistant bacteria. These changes influence the urobiome and the etiology of cystitis, thus indicating different treatment approaches for older versus younger females [23]. ABU is common in the elderly and treating it with antibiotics is no longer recommended due to both a lack of mortality benefit and harm of antibiotic overuse [24].

Innate immunity, barrier defenses, and the urobiome may play more of a role than specific immunity in protecting the urinary tract from infection [25]. The urinary tract, and especially the bladder, has several physical, chemotactic, and immune-mediated barriers to infection. The urogenital epithelium is turned over at a high rate relative to other tissue due in part to the regular passing of acidic urine as well as the local immune system’s normal function. At a normal pH of 5.5 or less, urine helps discourage bacterial growth, and mucosal tissue in the urinary tract produces organic acids to combat pathogens. Unlike the body’s typical Th1-dominant response to bacterial invasion, the uroepithelium produces a Th2-dominant response, allowing for quicker overturn of compromised uroepithelium but generating progressively weaker bactericide and bacteriostasis [26].

Thus, the syndrome of cystitis, whether or not it is accompanied by a bacterial infection, can have one or several compounding etiologies (e.g., infection, dysbiosis, uroepithelial dysfunction, or central sensitization). These can be difficult to assess accurately in the clinical setting, especially for chronic and recurrent cases. Antibiotics can address pathogen overgrowth and resulting inflammation; however, they may also exacerbate dysbiosis, drive resistance, or contribute to the imbalance of other systems.

### 1.5. The Urobiome

The urinary tract has a microbiome whose taxa is similar to that of the gut, with *Lactobacillus* being the most commonly occurring species [27,28,29]. Alterations in the urinary microbiota have been linked to non-infectious urologic diseases, such as neurogenic bladder dysfunction, interstitial cystitis, and urgency urinary incontinence [30]. Urine from patients with interstitial cystitis exhibits lower bacterial diversity and a higher abundance of *Lactobacillus* when compared to healthy controls [31]. These findings suggest that some cases of “suspected UTI” could be symptomatic expressions of microbial dysbiosis, which might only be exacerbated by antibiotics. Additionally, the use of low-dose, prophylactic antibiotics to prevent recurrent infection could further contribute to pathogenic resistance evolution by creating bacterial persister cells. Persister cells are genetically similar to previous generations but exhibit a greater fitness and virulence, which serves to optimize their permanent colonization in the indigenous microbial environment [32].

Advancements in metagenomics have allowed us to explore the gut microbiome as a potential reservoir for resistant pathogens. The high alpha diversity of microbiota within this ecosystem easily facilitates the horizontal transfer of antimicrobial resistance genes to susceptible pathogens. The concept of this “resistome” may offer more knowledge beyond the epidemiologic considerations of clinical isolates alone [33]. The intestinal microbiota composition varies between different niche environments along the length of the digestive tract, influencing metabolism, intestinal epithelial cell proliferation and permeability, and modulating the immune response, either directly or through crosstalk. As such, dysbiosis of the gut microbiome has been linked to a multitude of diseases beyond the intestinal environment, including many in the urinary tract. Moreover, microbial metabolites in the gut and other organs distal to the kidneys, bladder, and urethra are likely to influence the urinary microbiome and its homeostasis [34].

Metabolomic assay analysis suggests that species population may not be as important as the resultant metabolic pathways created [35]. Individual populations can differ in composition but perform the same functions to preserve environmental homeostasis. Recurrent UTI may be associated with an inability to reconstitute the normal microbiota, either as a result of long-term antibiotic use or other underlying host factors [36]. Ideally, microbial diversity allows for sustained essential metabolic activity within the microenvironment. A microbiome consisting of multiple species of bacteria with the ability to perform the same functions serves as an insurance policy against bacterial death due to pharmaceutical and environmental disturbances.

## 2. Whole-Person Approach to UTI

While up to 50% of uncomplicated cystitis cases spontaneously resolve within one week [37], medical attention and intervention are still important. In addition to the risk of worsening infection such as pyelonephritis, symptoms may take longer to clear without treatment and the length of time to symptom resolution is not tolerable for many patients [38]. Trials have compared ibuprofen alone to antibiotics and other therapies, hypothesizing that palliating symptoms will make the natural resolution of cystitis more tolerable and spare antibiotic usage [39,40]. While infection may resolve at similar rates when using ibuprofen compared to antibiotics, higher symptom burden and incidence of pyelonephritis are still consequences that require consideration [41].

These challenges elucidate the need for alternative therapies that provide symptomatic relief, have antipathogenic properties, and do not perpetuate recurrent infection. Investigations into the microbial diversity and population of the urobiome between healthy individuals and those with acute bacterial cystitis have prompted an interest in probiotic supplementation as a therapy for recurrent cystitis [42]. Several botanical and nutritional therapies, many of which originated in traditional medicine systems, have gained attention as promising alternative therapeutic models for cystitis. While most antibiotics employ a single pharmacologic mechanism to fight bacteria, botanicals generally contain a complex array of phytochemicals that have evolved in response to diverse threats, possibly yielding robustness against the development of targeted resistance [43]. Finally, urogenital hygiene and hydration are significant yet clinically undervalued contributors to urinary tract health that can be leveraged through behavioral interventions [44]. Altogether, these alternative approaches may be combined to target several mechanisms of action and address compounding etiologies for both acute and recurrent/chronic cystitis (see Table 1).

### 2.1. Probiotics

Lactobacilli are normal urogenital flora that can suppress uropathogens [45] and may have the potential to treat conditions of the urinary tract [46]. Various lactobacillus strains are essential to health; they help maintain a low pH, influence the immune system, and protect against infections. In particular, *Lactobacillus crispatus* has been associated with better bladder health when present in the urobiome [47] and *Lactobacillus* spp. Supplementation via vaginal suppository has been found beneficial in preventing UTI recurrence in women who are prone to recurrent UTI [48]. The oral administration of *Lactobacillus rhamnosus* GR-1 and *Lactobacillus reuteri* RC-14 restored favorable vaginal lactobacilli population profiles and reduced colonization by potentially pathogenic bacteria [49]. Microbial composition within the urobiome is integral to overall urological health and as such, restoration of a healthy urobiome is a valuable target in UTI therapy.

A 2015 Cochrane systematic review and meta-analysis of probiotics for UTI prevention found that, overall, probiotics alone did not seem significantly better than antibiotics or placebo in reducing the risk of UTI [50]. These results were less than conclusive due to a low number of participants and an unclear or high risk of bias among most studies. However, because the numerous beneficial functions of bacteria rely on complex interactions within the ecosystem, supplementation alone of a particular species may not be enough to restore and maintain balance to the urobiome [51]. It is possible that the administration of probiotics should be utilized in combination with other therapies to achieve the aims of pathogen clearance, urobiome restoration, and preventing future infection.

### 2.2. D-Mannose

D-mannose, a monosaccharide that is absorbed without metabolization and excreted entirely through the urine, has gained attention for its potential to prevent UTI by inhibiting bacterial adhesion to the uroepithelium. Mannosylated proteins in the bladder are one of the targets of lectin adhesion for *E. coli* and several other uropathogens; d-mannose competitively binds with uropathogens and promotes their clearance [52,53]. A systematic review and meta-analysis of clinical studies evaluating d-mannose for UTI prevention in adult women concluded that it is superior to placebo and possibly equally as effective as prophylactic antibiotics at preventing recurrence [54]. Additionally, d-mannose does not seem to contribute to resistance as it has no impact on *E. coli* metabolism, antibiotic activity, or bacteria viability [55], nor does d-mannose present an antibiotic-like activity, since it does not induce FimH variants that can modify bacterial adhesion [56]. A Cochrane review of the benefits and harms of d-mannose for UTI prevention is in process, as adverse events were infrequently reported in previous studies [57].

### 2.3. Cranberry (Vaccinium macrocarpon)

The biochemical constituents of cranberry that rationalize its use in UTI include several bacteriostatic acids (quinic, malic, citric) and proanthocyanins that, similar to d-mannose (also present in cranberry), inhibit the adhesion of type 1 fimbriae on *E. coli* [58]. Additionally, studies have suggested that daily consumption of cranberry can have anti-inflammatory and antioxidant effects on the bladder [59,60] and boost beneficial gut bacteria species such as *Bifidobacterium longum* and *Akkermansia muciniphila,* which inhibit the growth of pathogenic organisms and reduce intestinal inflammation, respectively [61,62]. One study found that participants who consumed cranberry had significantly less abundant levels of the *Flavonifractor* species (OTU41) compared to the placebo group [63].

Previous research has shown an association between high host populations of OTU41 and a variety of mental health disorders, autoimmune diseases, and obesity [64,65]. OTU41 is involved in the transport of tryptophan and cobalamin, thus it is a key regulator of the gut microbiome and microbial interactions within niche environments [66,67]. The pathogenesis of *E. coli* colonization requires the biosynthesis of the metabolite ethanolamine (EA) in the presence of cobalamin. EA acts as a source of nitrogen to outcompete other microbes and increases the expression of virulence genes [68]. Furthermore, elevated levels of tryptophan metabolites have been associated with acute bacterial cystitis in pediatric patients, suggesting tryptophan metabolism could be an important factor in recurrent UTI [69].

A recent systematic review and meta-analysis of randomized-controlled trials on healthy, nonpregnant adult females with a history of UTI found cranberry to be effective in preventing UTI recurrence compared to placebo [70]. Another trial published in 2019 tested a combination of cranberry and propolis against placebo for women with recurrent UTI and found it to be significantly better than placebo with fewer UTI occurrences in the 3-month period following treatment and a longer time to first UTI recurrence [71].

Further examination of cranberry and its role in modulating the gut microbiome, including possible effects on the abundance of the *Flavonifractor* species and its influence on *E. coli* pathogenesis, is needed to characterize proper therapeutic use in the treatment of recurrent UTI.

### 2.4. Arctostaphylos Uva-Ursi

Arbutin is a glycoside derived from extracts of the leaves of *Arctostaphylos* uva-ursi, which has been used traditionally for the treatment of UTI in Europe, America, and Asia [72]. β-arbutin exhibits antimicrobial activity and has been shown to destroy a variety of both Gram-negative and Gram-positive bacteria including *Staphylococcus aureus, Enterococcus faecalis*, and *Escherichia coli,* as well as the antibiotic-resistant strains: Escherichia coli ESBL R194, *Enterococcus faecalis* HLAR, and *S*. *aureus* MRSA K31 [73,74]. β-arbutin has also shown hepatic anti-inflammatory effects as well as restorative histopathological changes in the liver, pancreas, and kidneys damaged by diabetes in rats [75,76]. While there is a well-documented concern that hydroquinone, a derivative of arbutin, could be nephrotoxic, a study on rats did not find any damage to kidney function or negative effects on the integrity of the DNA in kidney cells [77]. Additionally, an in vitro safety assessment of the strawberry tree (*Arbutus unedo* L.) water leaf extract and arbutin in human peripheral blood lymphocytes showed no toxicity or cellular damage [78]. A randomized controlled clinical trial of women aged 15 to 75 years with a history of recurrent, uncomplicated cystitis found that oral administration of a combination of arbutin, birch, berberine, and forskolin in conjunction with D-mannose reduced the incidence of recurrent episodes of cystitis and positive urine cultures during treatment and at follow-up [79]. While few human studies have been performed on uva-ursi, the existing preliminary research shows promising anti-microbial and anti-inflammatory synergistic effects worthy of further investigation.

### 2.5. The Vaginal Microbiome and Behavioral Considerations

Sexual health and the vaginal microbiome also play a key role in UTI recurrence. While intestinal microbiota is the ultimate source of bacterial strains causing cystitis and pyelonephritis in the majority of cases, changes in the characteristics of the vaginal microbiota (particularly the loss of the normally protective *Lactobacillus* spp.) increase the risk of UTI [80]. Alterations in the vaginal microbiome may result from several influences such as estrogen deficiency, the use of antibiotics, certain contraceptives, or other causes [81,82].

Bacteria colonize the vaginal introitus and periurethra and can ascend via the urethra to the bladder or kidneys to cause further infection [93]. Studies have shown that women with recurring UTI often have increased rates of colonization with *E. coli* along with a depletion of H_2_O_2_ producing lactobacilli, and women with recurrent UTI who lacked vaginal H_2_O_2_-producing lactobacilli had a 5-fold increased risk of *E. coli* vaginal colonization compared to women with H_2_O_2_-producing lactobacilli [94,95]. These findings suggest a protective relationship between H_2_O_2_-producing lactobacilli and preventing *E. coli* vaginal colonization and hint at the complexity of the effects of the microbiota of distal environments on the pathogenesis of UTI.

Hydration and voiding habits play key roles in clearing and preventing bacterial colonization of the urinary tract. Infrequent voiding and dehydration are widely recognized risk factors for UTI, and increasing water intake can reduce the recurrence of UTI in otherwise healthy but highly susceptible females [83]. However, recommendations of maintaining proper hydration and educating patients on vaginal health are often neglected components of clinical guidelines for treating uncomplicated cystitis.

Diet may also play an important role in UTI prevention and treatment. A diet high in polyphenolic compounds obtained from plants, especially berries rich in proanthocyanidins, is likely to be protective from uropathogens [84,85]. Furthermore, strains of uropathogenic *E. coli* isolated from urine in UTI patients have been found to be different from any human intestinal strains, but nearly identical to pathogenic *E. coli* found in pork and poultry [86,87]. One observational study showed that middle-aged to elderly UTI patients who ate high amounts of pork and poultry products had more drug-resistant strains of *E. coli* than those who ate less [88]. Conversely, a Taiwanese cohort study demonstrated that a vegetarian diet was associated with less occurrence of uncomplicated UTI compared to a non-vegetarian diet, particularly in female non–smokers [86].

Plant-based fibers and carbohydrates play a primary role in supporting diversity of the gut microbiome, which is associated with a host of positive influences on multiple organ systems, including the urinary tract [89,90,91]. The American Gut Project, a study of the gut flora composition of more than 10,000 average citizens, found a direct correlation between the number of different plants incorporated into a person’s diet and the diversity of their gut microbiota [92]. Participants who consumed 30 or more different types of plant foods per week had significantly more diverse microbiomes than those that consumed fewer than 10 types of plants, regardless of adherence to varied specialized diets.

While there is more to be discovered regarding the downstream impact of diet on the urobiome and UTI occurrence, and in light of its myriad of other health benefits that are well-established, clinicians can maintain a low threshold in recommending a high plant-based fiber diet for patients at risk of UTI. Behavioral modification in regard to hygiene, hydration, and diet should be a mainstay of antibiotic-sparing strategies to UTI management. Motivational interviewing is an effective strategy for fostering health-promoting behaviors such as diet and physical activity and could be highly serviceable as an adjunct to non-pharmacologic therapy for cystitis [96,97,98].

### 2.6. Current Research on Multimodal Botanical Therapies

Formulations of multiple botanicals for treating urinary tract disorders are common in several whole systems of medicine, including Naturopathy [99], Ayurveda [100], and Traditional Chinese Medicine [101,102]. Although the biochemical complexity of herbs compared to pharmaceuticals poses some difficulty to scientific study, new technologies are increasingly capable of understanding the interplay of systems (e.g., deep-learning models and combining ultra-high-performance liquid chromatography with tandem mass spectrometry) [103,104,105]. Certain combinations of herbs found in traditional formulas have synergistic qualities by producing novel compounds or potentiating the effects of desired active compounds from each herb [106].

One study evaluated a combination of d-mannose plus herbs with pre-clinical evidence of anti-infectious action (uva-ursi, *Betula pendula*, and *Berberis vulgaris* dry extracts, with or without the addition of *Coleus forskohlii* as an adjuvant) and compared them to a control combination of proanthocyanins plus d-mannose in a three-arm randomized trial [54]. The herbal intervention groups had fewer episodes of UTI recurrence and less bacteriuria than the control group during the intervention period and at 24-week follow-up. A similar approach was found to be successful in a pilot study of 85 patients with recurrent UTI. Utilizing an attack dose of 1000 mg d-mannose and 200 mg dry willow extract (salicin) three times daily for three days followed by maintenance treatment with 15 days of d-mannose and *L. acidophilus* (La-14) each month for two consecutive months showed a significant reduction in the symptomology of bacterial UTIs [107].

A high-quality non-inferiority study from 51 sites across Europe compared a commercially available combination of herbs (﻿BNO 1045—*Centaurii herba*, ﻿*Levistici radix*, ﻿*Rosmarini folium*) to Fosfomycin or placebo [108]. Females aged 18–70 who presented acute uncomplicated UTI were randomized 1:1:1 to each trial arm. The clinical success rate for the herbal combination (84%) compared to Fosfomycin (90%) met the predetermined margin of noninferiority with clinical success being defined both by syndrome resolution and sparing of antibiotic use during the 38-day observation period. These results argue strongly that otherwise healthy patients with uncomplicated UTIs can be treated safely with this botanical compound as a first-line strategy.

### 2.7. Vaccine for UTI

Vaccination is a public health measure used for infection, and as such vaccines are being developed for UTIs. The overgrowth of many bacteria can cause UTIs including *Escherichia coli*, *Klebsiella pneumoniae*, *Proteus mirabilis*, *Enterococcus faecalis,* and *Staphylococcus saprophyticus*. Uropathogenic *E. coli* is the most common microbe leading to UTI and this is the microbe targeted by most vaccines in development [109].

There are significant challenges to developing vaccines for UTIs. Importantly, UTI vaccines target bacteria that are normal microbiota, and mounting an immune response against commensal bacteria is likely to generate significant side-effects. To address this challenge, researchers are developing UTI vaccines against virulence factors (purified protein) instead of whole microbes. While this strategy may reduce side effects, it may also not lead to long-term immunity, which counters the intention of vaccination [110].

There are multiple strains of pathogenic *E. coli*; a single strain vaccine may not be effective against all strains. Furthermore, as discussed previously, UTIs can be caused by multiple microbes. Thus, a vaccine against *E. coli* will not protect against *Klebsiella, Proteus*, or other pathogenic urinary bacteria. While vaccines provide an interesting approach, the challenges may outweigh the benefits. Karam et al. provides a more complete discussion of UTI vaccines [109].

## 3. Future Directions

A whole-person, multi-modal approach to uncomplicated UTI may be more successful than any individual therapy alone. Many other botanicals and naturally derived substances not mentioned above demonstrate antimicrobial activity in preclinical studies, but relatively few have been evaluated in human trials [111]. Future research should aim at reproducing and scaling up clinical studies of botanical compounds that show promise for acute and recurrent UTI. Strong, pragmatic study designs are needed to identify relative best practices for routine use of non-antibiotic UTI therapies and their tolerability for patients.

Considering the public health impact of antibiotic resistance (and in light of the clinical evidence for non-pharmaceutical therapies), we argue for the support of dissemination and implementation of these therapies for uncomplicated UTI in the appropriate clinical context. The NCCIH Strategic Plan FY 2021-2025 emphasizes the need for multicomponent interventions with the aim to holistically improve patient outcomes rather than focusing only on disease management. Complementary and integrative health approaches should engage multiple therapeutic systems, and clinical research should not shy away from a full investigation of the mechanistic complexities of the interconnection between various physiological systems.

As the clinical evidence aggregated herein suggests, a strategy of attempting multimodal therapy prior to prescribing antibiotics is safe and could simultaneously achieve the aims of syndrome resolution, microbiome preservation, and antibiotic sparing.

## 4. Conclusions

Most urinary tract infections are treated in the outpatient setting with antibiotics, which significantly drives antibiotic resistance. Recent understanding of the urinary tract microbiome and the role of the innate immune response suggests that therapy should target functional restoration to support the host response in combating bacterial invasion and discouraging recurrent infection. Several novel approaches for the treatment and prevention of cystitis (including probiotics and botanicals) have been proposed and tested. The benefit of current antibiotic prescribing patterns for cystitis no longer outweighs the known harms. Non-pharmacologic strategies for uncomplicated UTI deserve more attention and resources for their successful implementation into clinical practice.

## Figures and Tables

**Table 1 antibiotics-11-00218-t001:** Whole-Person Approach to Uncomplicated UTI and Urobiome Health.

Therapy	Strategy	Mechanism	References
*Lactobacillus* spp. (Probiotics)	Decrease pathogenic bacteria by competing with pathogenic microbes for nutrients and space; May provide healthy microbial metabolites.	Urobiome support Anti-adhesion Anti-inflammatory	[45,46,47,48,49,50,51]
D-mannose	Inhibits pathological bacterial adhesion to uro-epithelium.	Anti-adhesion	[52,53,54,55,56,57]
Cranberry	Increases growth of beneficial bacteria and decreases growth of pathogenic bacteria.	Urobiome support Anti-adhesion Anti-septic Anti-inflammatory Diuresis	[58,59,60,61,62,63,64,65,66,67,68,69,70,71]
Arctostaphylos uva-ursi	Increases urine flow to decrease pathogenic bacteria.	Anti-septic Anti-inflammatory Diuresis	[72,73,74,75,76,77,78,79]
Sexual hygiene & Contraception choice	Slow contamination.	Anti-contamination pH level maintenance	[80,81,82]
Hydration	Increases urine flow to clear pathogenic bacteria.	Urobiome support Anti-contamination Diuresis pH level maintenance	[83]
Diet	Increase growth of beneficial bacteria; slow contamination; decrease growth and adhesion of pathogens	Anti-contamination Urobiome support Anti-inflammatory Anti-adhesion	[84,85,86,87,88,89,90,91,92]

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
