# Peer review of "Whole-Person, Urobiome-Centric Therapy for Uncomplicated Urinary Tract Infection"

_antibiotics, 2022, doi:10.3390/antibiotics11020218_

Round 1

Reviewer 1 Report

Garofalo et al presented a Review on Urobiome-centric therapy for uncomplicated UTI. The Review is well structured and covers most more or less known approaches to treatment and preventention of uncomplicated UTIs without antibiotics. It's concise and has a strong focus on "integrative" medicine which is a field of medicine that is often looked down upon by clinicians.

Specific comments:

1. I don't understand the title. Why exactly is it called "whole person?" this appears to be a phrase of integrative medicine. I would recommend changing the title to "Urobiome-centric..." 

2. I am unsure whether the strong focus on Arctostaphylos uva-ursi is justified by the fact that there is a single RCT that showed beneficiary outcome (in combination with Mannose - which may be the confounder to this outcome?). Maybe the authors can explain why they dedicated more space to this topic than to mannose, where apparently more data exists.

3. A topic that I would ask the authors to go into detail is "vaccination against UTI". I think it is a topic worthwile to include in this review and add to its content given the potential "manipulation" of the microbiome.

4. Is there any data on vegan / vegetarian diet? Could you please comment on - and potentially integrate on the study by Yen-Chang Chen et al (The risk of urinary tract infection in vegetarians and non-vegetarians: a prospective study DOI: 10.1038/s41598-020-58006-6 )

5. Figure 1: I think Figure one could be redone to be both more appealing and easier to understand. Visualization could help the reader to memorize the mechanisms of the graph and the different mechanisms of prevention.

6. Defintions: while it may common sense to most readers the manuscripts lacks clinical defintions of bacteriuria, uncomplicated uti and complicated uti. additionally most studies were done on female, given that a lot of people still consider the male UTI to be always "complicated".

7. I would love to see if the authors could go into details on the differences between males and females (it appears that most studies were done on females)

Author Response

Thank you sincerely for your thoughtful review. Our responses are as below:

 1) We used the term 'whole-person health' because this is the terminology being used by the NCCIH, which funds us. We included some language and a citation to direct the reader to explore more about this emerging field. The term is inconsequential to the substance of the paper and can be removed if you think it necessary.  2) The reason there is more writing under the uva-ursi section is because the discussion is more nuanced and complex than the research and mechanism of mannose, which is much more straightforward. Uva-ursi is more popular in herbal remedies marketed to the public than mannose is, even though there are more studies done on the latter. Heather agreed with my suggestion that we combine the sections and lead into the multi-modal overlap so that our mention of mannose doesn't seem so brief in comparison. 3) We included a section on vaccines as suggested and its relative considerations within the scope of this review. 4) We have included the excellent resource you shared related to dietary approaches in UTI management and expanded upon this aspect. 5) The figure has been adapted for clarity. 6) We have added introductory sections on prevalence, pathophysiology, and diagnostic criteria. 7) We added language explaining our focus on females, being due to the historical restriction of the definition of "uncomplicated" UTI and the impact this has on the literature.   We're grateful for your suggested improvements and will be eager to read your response.

Reviewer 2 Report

This is an interesting manuscript. The authors have to address the following before it can be considered.

  1. The definition, diagnostic criteria, and prevalence of UTI need to be discussed.
  2. The classification and pathophysiology of UTI also need to be discussed.
  3. There are two commas before botanicals, please delete them (page 4, line 155).
  4. Move the period after ref.58 and ref.59 (page 6, line 225-226).
  5. After a duplication-check on your manuscript of antibiotics-1561883, there a few paragraphs/sentences are almost the same with the published papers (e.g., lines 32-34, 36-44, 244-274), could you please rewrite/make revisions accordingly?
  6. The reference style needs to be consistent.

Author Response

Thank you sincerely for your thoughtful review.

Introductory sections with definitions, prevalence, diagnostic criteria, classification, and pathophysiology were added. Typos and wording to the sections you outlined have been corrected. References have been uniformly formatted.

We're grateful for the suggested improvements and will look forward to reading your response.

Round 2

Reviewer 1 Report

Garofalo et al presented a review on urobiome-centric therapy and prevention of uncomplicated UTIs. They incorporated my suggestions to my content and I think this is an easy to read and comprehensive review on this newer field of medicine. Thank you for implementing the sections on Diet and Vaccination as well as the definitions.